# Recent Trends in Multiclass Analysis of Emerging Endocrine Disrupting Contaminants (EDCs) in Drinking Water

**DOI:** 10.3390/molecules27248835

**Published:** 2022-12-13

**Authors:** Abigail Lazofsky, Brian Buckley

**Affiliations:** Environmental and Occupational Health Sciences Institute, Rutgers University, 170 Frelinghuysen Road, Piscataway, NJ 08854, USA

**Keywords:** contaminants of emerging concern, analytical methods, instrumentation, evaluation, bioassay, non-targeted analysis

## Abstract

Ingestion of water is a major route of human exposure to environmental contaminants. There have been numerous studies exploring the different compounds present in drinking water, with recent attention drawn to a new class of emerging contaminants: endocrine-disrupting compounds (EDCs). EDCs encompass a broad range of physio-chemically diverse compounds; from naturally occurring to manmade. Environmentally, EDCs are found as mixtures containing multiple classes at trace amounts. Human exposure to EDCs, even at low concentrations, is known to lead to adverse health effects. Therefore, the ability to evaluate EDC contamination with a high degree of sensitivity and accuracy is of the utmost importance. This review includes (i) discussion on the perceived and actual risks associated with EDC exposure (ii) regulatory actions that look to limit EDC contamination (iii) analytical methods, including sample preparation, instrumentation and bioassays that have been advanced and employed for multiclass EDC identification and quantitation.

## 1. Introduction

Contaminants of emerging concern (CECs) are defined as chemicals that were not previously identified or have new environmental health or toxicological understanding [1]. There is a wealth of chemicals falling under this umbrella that are worthy of further investigation. For the purposes of this review, we narrow the scope of focus to chemicals with a shared toxicological endpoint: endocrine disrupting compounds (EDCs). Like many CECs, a majority of EDCs share common characteristics of a potential for long-range transport, a high persistence in the environment, a high likelihood for bioaccumulation, and a concerning toxicity [1]. Furthermore, we anticipate the number of environmental contaminants that have endocrine disrupting properties to increase as manufacturing of novel compounds continues to grow. In this review, we will discuss the actual and perceived health risks that accompany exposure to EDC mixtures, the occurrence of EDCs in our potable water system, and the current analytical strategies and methodologies that are being utilized for simultaneous analysis of multiple classes of EDCs in environmental water.

## 2. Emerging Unregulated Contaminants: Endocrine Disrupting Compounds

Endocrine disrupting compounds (EDCs) are both natural or synthetic chemicals that can lead to this internal imbalance, disrupting our hormones’ regulatory pathways affecting development, growth, reproduction, metabolism, immunity, and behavior [2]. The endocrine system includes the hypothalamus, parathyroid, thymus, thyroid, pancreas, adrenal, pituitary, and ovaries/testes. Together, these glands are responsible for maintaining hormone levels, effectively controlling and balancing every major system in our body [3]. Hormone signaling is a sensitive process that can easily become unbalanced with the introduction of various stressors. EDCs have several mechanistic approaches that result in hormone disturbance: (1) mimicking endogenous hormones (2) binding to the receptor and blocking the desired hormone from attaching, and (3) interfering with the way natural hormones or their corresponding receptors are made or controlled [4]. There is a heightened concern associated with EDC contact in more susceptible populations, such as pregnant mothers and children [2]. These compounds can cross the placental barrier, have been identified in breast milk, and play a highly influential role upon exposure during key childhood developmental windows. EDCs have been reported to be found ubiquitously within the population [5], further highlighting the importance of understanding their behavior in the environment and the body so that risk mitigation strategies can be designed to lessen their deleterious effects. 

## 3. Human Health Risk Assessment and Exposure to EDCs

EDCs are a group of diverse chemicals, often present in environmental matrices in trace amounts. Because humans are regularly exposed to a mixture of physio-chemically diverse compounds rather than a single contaminant, predicting the severity of health consequences remains complex. In fact, until recently, risks from chemical mixtures were considered negligible as long as exposures to all individual chemicals within the mixture were below the levels deemed safe for each chemical alone [6]. However, there is an ever-growing body of evidence that low concentrations of xenoestrogen blends can have additive or potentiating effects [7,8]. Remaining challenges surrounding the impacts of EDC exposure include incomplete data for newly manufactured chemicals (i.e., the toxicity of individual compounds), unknown combinatory effects, accurate measurement of environmental concentrations, and relevance of exposure levels [9,10]. Furthermore, the full impact of human exposure to EDCs is difficult to study because adverse effects manifest at later ages, as latency between early exposures to EDCs and adult disease could be more than 50 years [11]. 

Many investigations and in-depth reviews summarizing the risk of individual EDCs or compound classes have been published [12,13,14,15,16,17], while reports of multiclass environmental mixtures remain limited [18,19,20]. Fan et al., 2021 quantified 10 EDCs of high concern in sediment and surface water samples collected from the Jiangsu Province [21]. Based on calculated risk quotients, 4 compounds (bisphenol-A (BPA), nonylphenol, (2-ethylhexyl) phthalate, and 4t-octylphenol) were determined to be medium to high risk compounds within the examined environmental matrices; however, they do not address their cumulative risk potential. Another study relied on epidemiological associations, analyzing exposures to common EDCs and observed health outcomes within a cohort of human-child pairs, concluding that exposures to EDC mixtures was associated with increased risk of offspring developing a language delay [22]. Additional studies emphasize the importance of how chemicals should be grouped within mixtures, highlighting their similarities (i.e., mode of action, health outcomes, etc.) for effective risk assessment [23,24]. 

## 4. Breadth of Compounds (Classes) Categorized as EDCs

There are many compounds that can be categorized as EDCs, from the naturally generated (e.g., zearalenone mycoestrogens), the hybrid natural and synthetically formed (e.g., polyaromatic hydrocarbons (PAHs)), the industrial (e.g., bisphenol analogues, per- and polyfluoroalkylated substances (PFAS)), and the cosmetic (e.g., personal care products like triclosan). The number of compounds exhibiting endocrine disrupting properties is continuously growing, as more potential EDCs are introduced in an effort to replace currently existing ones (e.g., short-chains PFAS as substitutes for long-chain PFAS). It is worth acknowledging that there are too many EDCs to succinctly summarize within a single review, while understanding the significance of discussing the overall importance of EDC analysis. Most methods discussed in this review include, but are not limited to, plasticizers (e.g., BPA, PBDEs), hormones, alkylphenols, PFAS, flame retardants, and pesticides. It is worth noting that PFAS have previously been shown to display endocrine disrupting activity [25,26], and have been recognized by the US Environmental Protection Agency (EPA) as exhibiting endocrine disruptive health effects, such as decreased fertility, developmental delays, and hormonal interference [27].

Because of the variety of different structural classes included under the umbrella of EDCs, the routes of exposure are diverse; they can be inhaled through the ambient air, ingested via contaminated food and water, or absorbed dermally. For the purpose of this review, the focus will be on ingestion of water, as it remains the major exposure route for EDCs [28]. Environmentally, EDCs can get into the water supply from numerous sources such as treatment plant effluent, human and livestock excretion, manufacturing processes, and environmental runoff (Figure 1) [29]. Previous research has highlighted the presence of many classes of EDCs in several areas along our potable water system in drinking water [30,31], surface water [31,32], groundwater [33], and wastewater [34]. Finding these contaminants in multiple water sources demonstrates that safeguards designed to prevent.

EDCs from contaminating water have not been effective. There are numerous recently published reviews discussing the environmental fate, transport and occurrence of EDCs in depth, with focus on specific emerging compounds classes such as PFAS, alkylphenolic compounds, personal care products (PCPs), and brominated flame retardants (BFRs) [35,36,37]. Recognition of the presence of these compounds in water sources poses the question about whether they have a net effect on environmental health. While not the principal focus of this review, assessing the environmental impact of these contaminants is the driving force behind the creation of analytical methods to identify and quantify these compounds.

## 5. Government Regulations: United States

The United States has implemented several regulatory and non-mandatory guidelines to defend the integrity of drinking water sources. The Safe Drinking Water Act, passed in 1974, seeks to protect public health through the regulation of the public drinking water supply from source to tap [38]. As a result, the US EPA is given the authority to set national health-based standards for drinking water. These standards, known as National Primary Drinking Water (NPDW) Regulations, are legally enforceable maximum concentration levels (MCLs) and treatment techniques that must be maintained by public water systems [39]. It is worth noting that the US EPA can also define maximum contaminant level goals (MCLGs) based on predicted adverse health effects, but they are non-enforceable and act more as guidelines for public water utilities [40]. Regulated compounds include those that fall under the broad categories of microorganisms, disinfectants and their byproducts, inorganic chemicals, organic chemicals, and radionuclides. However, this list only covers 52 organic chemicals and 1 organic compound class (polychlorinated biphenyls, PCBs), a small percentage compared to the breadth of compounds that have previously been identified in water sources. 

The US EPA publishes a Contaminant Candidate List (CCL) every 5 years that includes compounds not currently covered by NPDW regulations but are expected to be found in public water systems. The most recently published CCL 4 contained 97 chemicals and contaminant groups and 12 microbial contaminants; a draft of the fifth iteration was released in July 2021 and consists of 12 microbial contaminants and 66 chemicals including emerging per- and polyfluoroalkylated substances (PFAS) [41]. The Unregulated Contaminant Monitoring Rule (UCMR) is another SDWA provision that looks to cover those compounds not considered to be regulated by NPDW regulations. The UCMR is used to collect data for those contaminants that are suspected to be present in drinking water and do not have set health-based standards, many of which are CECs [42]. To address the growing concerns for EDCs specifically, the US EPA developed the Endocrine Disruptor Screening Program (EDSP), a two-tiered screening approach that analyzes environmental contaminants for their potential effect on our hormonal systems [43]. It utilizes both in vitro and in vivo assays to not only identify potential EDCs, but to determine adverse health effects, dose response relationships, and perform risk assessments [44]. Nearly 20 years after its conception, the EDSP has only published 2 lists of chemicals identified as possible EDCs: 67 identified in 2009, and 134 identified in 2010. Chemical regulation is highly based on the risk assessment of individual chemicals or compound classes. With an estimated 84,000 chemicals reported to be in commerce in the United States alone [45], the current government regulations will continue to overlook the real impact that complex EDC mixtures play on environmental and human health.

## 6. Public Health Perception of Drinking Water Quality and EDCs

Public trust in the safety and quality of potable water is key for effective risk mitigation and maintaining open communication between regulatory agencies and citizens who are directly impacted by their decisions. Under the SDWA, community water systems are required to publish Consumer Confidence Reports, which provide local water users information about the quality and status of their tap water, establishing a base level of trust between consumers and water utility managers [46]. Despite the mandated availability of water quality information, this data is not necessarily making it back to most consumers, as a study conducted in 2021 reported that 61% of participants (n = 352) had little to no information about their neighborhood water quality [47]. A 2015 study of 1200 adults found that 56% expressed concerns over the quality of their household water supply, with nearly 60% expressing a major concern related to waterborne contaminants [48]. A study conducted by Javidi and Pierce confirmed previous findings that low-income minority groups disproportionally perceive their drinking water as unsafe, often opting for bottled water alternatives [49]. Overall, a low awareness surrounding water source protection, environmental and health risk assessment, and water treatment efficiency is emphasized by the differences between public risk perception and reported drinking water quality [29,50].

Past studies exploring the public perception of EDCs have either focused on specific demographic groups (i.e., pregnant women), health effects (i.e., infertility) or individual compounds (i.e., BPA). Interestingly, a cohort of pregnant women had reportedly low concern with personal susceptibility to EDC exposure while acknowledging their belief that exposure to EDCs was dangerous, highlighting the discrepancies in public EDC education [51]. This knowledge gap is only exacerbated when exploring the extent to which EDCs in drinking water are viewed as a serious health risk by the public. A study conducted by Wee et al., 2022 looked at how a population in Malaysia relying on tap water with potential EDC contamination perceived their safety and risk associated with water consumption [52]. It was reported that 85% of their study population (n = 140) was highly concerned about their tap water quality; however, it was noted that this concern was not likely attributed to the presences of EDCs. Similar conclusions are drawn in Kelly et al., 2020, where interviews revealed participants felt there were more pressing issues, like climate change and obesity, that took higher priority than their EDC exposure [53]. Additionally, both surveyed populations believed their government had the ability to regulate any EDCs present in their drinking water supply despite the fact most EDCs present were not covered under their respective guidelines. From these studies, it can be concluded that much of the general population are unaware of what EDCs are and how their presence can impact their health and their progeny. 

Increased communication between regulating bodies and the public is a key step in preparing individuals to handle exposure scenarios, as they will continue to increase as manufacturing and the population expand. A primary step in effective risk communication of EDCs is accurate identification and quantification of harmful chemicals present in drinking water sources. The remainder of this review summarizes the analytical methods used in the identification and quantification of multiclass EDCs in water sources from recent publications (2009–2022). Using the key phrases “multiclass analysis endocrine disruptors” and “analysis method endocrine disruptors drinking water”, the literature search identified 127 articles (PubMed), with 112 remaining after previous review papers were removed. The articles were then manually assessed by title and abstract within Endnote, and studies without major discussion of analytical methodologies were eliminated. Of the remaining articles (n = 54), studies examining matrices other than water (e.g., urine, dust, etc.) were excluded (n = 12). Additionally, only papers discussing at least 2 classes of compounds qualifying as EDCs were included, for a total of 34 studies reviewed for discussion. 

## 7. Analytical Methods

### 7.1. Targeted Analysis

Generally, targeted analysis aims to detect a number of known compounds using specific analytical procedures. Similarly, approaches to large-scale EDC analysis in water have traditionally relied on targeted lists of chemicals within the same compound class. There have been several review papers already published exploring the analysis of specific compound classes, like pesticides and flame retardants, belonging to the EDC category [54,55,56,57]. Multiclass EDC analysis is more complex, as the targeting of a variety of physio-chemically different compounds within a single method can lead to challenges maintaining sensitivity. However, use of single, cohesive analytical methods allows for a broader range of chemicals to be examined while simultaneously cutting the need for multiple techniques, which in-turn, decreases the overall time and cost of analysis. 

### 7.2. Non-Target Analysis (NTA)

As new environmental contaminants emerge, analytical labs recognize that they are often playing catchup with method development when a new regulation (MCL) is issued for compounds that were previously unregulated and therefore, not targeted by existing standard assays. To increase the utility of many broad-based analytical techniques, there is an on-going migration from targeted methodologies to non-targeted analysis (NTA) of environmental contaminants, allowing for additional identification, and possible quantitation, of unknown compounds. Additionally, some environmental/government agencies, like the EPA, recognize the need to know what is in the water even though many chemicals are currently unregulated [58]. Sometimes this is in response to secondary markers, such as a population of sentinel species showing more than expected numbers of females [59].

Difficulty remains in trying to focus such a broad approach to compound analysis in a way that keeps public health concerns in mind as non-targeted analysis is generally non-quantitative, occasionally semi-quantitative [60], and their results are often dominated by compounds with few, if any, health effects. Analysts that have studied compounds with similarities in toxicological endpoints utilize a combination of techniques, such as fractionation and bioassay analysis, to further help with separation and identification of desired contaminants (discussed more later). NTA seeks to circumvent multiple assay scenarios my measuring “all” of the potential contaminants. However, the most challenging aspect of NTA is the lack of uniformity; there are no standardized programs that everyone can use, making determining a workflow very difficult. 

## 8. Sample Preparation

As stated previously, detection of EDCs in water is likely to occur at the trace level. Because of the non-uniform nature as well as the wide range of contaminants anticipated to contribute to the overall pollution of environmental matrices, pretreatment of collected water samples is a necessary step prior to the analysis of complex water mixtures. Proper sample preparation can help decrease matrix interferences and contribute to better analytical sensitivity. The most common method used for environmental water analysis is solid phase extraction (SPE). SPE allows for proficient extraction, cleanup, and concentration of analytes, improving recovery yields in a shortened amount of preparation time [61]. When a water sample is passed through an SPE column, adsorbent packing material retains the analytes of interest until they are eluted in a suitable solvent. Traditionally, analysis of single compound classes allows for proficient selection of the most optimal column packing type; however, when expanding the number of compounds included for simultaneous analysis, a more generalized column helps prevent the unintentional exclusion of compounds of interest. There are three kinds of SPE cartridges: (1) normal phase (used for polar analytes), (2) reverse phase (used for polar to non-polar analytes), (3) ion exchange (used for charged analytes) [62]. EDCs can be nonpolar (i.e., polybrominated diphenyl ethers (PBDEs), polycyclic aromatic hydrocarbons (PAHs)); however many EDCs are more polar than traditional contaminants [63]; therefore, reverse-phase columns are most often used for multi-class analysis. Studies utilizing this method select manufacturer brands containing various polymeric compounds, including polystyrene-divinylbenzene and/or vinyl pyrrolidone polymers [64,65,66], C18-bonded polymers [67] or manufacturer-trademarked polymers [68]. Multiple sorbents can be utilized for more proficient extraction of a wide range of physio-chemically diverse compounds. One study analyzing CECs in wastewater combined 4 different SPE-packing materials within a single in-house cartridge to achieve sufficient enrichment and extraction of their targeted analytes [69]. SPE-techniques can be performed either manually or automatically with on-line SPE techniques. On-line treatment allows for the consecutive extraction and instrument analysis of samples through the automatic adjustment of intake valves rather than manual transition from SPE column collection to instrument sample loading. Goeury et al., 2019 optimized an on-line SPE protocol using a set of two Hypersil Gold C18 columns in succession, resulting in extraction efficiencies ranging between 64–79% for 13 EPA-priority hormones and BPA [70]. An on-line approach can provide advantages in the overall need for materials, time of analysis, and reproducibility [71]. 

Liquid-liquid extraction (LLE) is another preparation method that has been used for analyzing EDCs in water samples. LLE employs two immiscible liquids that are mixed together, allowing for compounds to partition out and dissolve based on their relative solubilities. Previous literature examining phthalates and BPA in various source waters extracted compounds using salt-assisted LLE, where dichloromethane was found to successfully extract all 8 analytes [72]. Another study examining phthalates, BPA, and 4-nonylphenol also used dichloromethane to successfully collect their analytes from drinking water after separately adjusting and extracting their samples in both acidic and basic conditions [73]. Overall, LLE is not chosen often for multiclass methods when compared to techniques like SPE. This is likely due to the expected differences in the polarity of multi-class EDCs, as well as the large volumes of solvent and time required to perform effective targeted extraction. 

Another extraction method that is favorable for environmental water analysis is solid phase microextraction (SPME). SPME is a field ready, solvent-free technology that employs a sorbent-covered fiber straight into the sampling matrix, where analytes are adsorbed until equilibrium is reached [74]. After collection, SPME fibers can be directly transferred to analytical instrumentation for analysis. Similar to SPE column packing materials, the coating on the SPME fiber can be selected to either specifically capture a single class of compounds or broadly analyze a variety of compound classes simultaneously. Due to its ease of use, portability, and selectivity, SPME is an increasingly utilized approach for analyses of various water matrices. A review published in 2016 describes the use of SPME for the analysis of water samples and provides a more in-depth examination into its comparison to traditional extraction methodologies as well as validated methods used for analysis [75]. SPME techniques have been used in the analysis of individual compounds and classes of EDCs like BPA [76], 4-nonylphenol [77], pharmaceuticals [78], and pesticides [79]. Currently, its utilization in multiclass EDC analysis is not as prominent. A recent study published by Alimzhanova et al., 2022 describes the miniaturization of SPME for the analysis of 8 EDCs (5 hormones, 1 herbicide, 2 pharmaceuticals) in drinking water using a fiber coated with a combination of divinylbenzene/carboxen/polydimethylsiloxane, allowing for analysis of low-ppb concentration analyte levels [80]. 

## 9. Instrumentation

The ability to study emerging contaminants in general and, more specifically, EDCs, has been facilitated by the evolution of measurement technologies. In particular, gas or liquid chromatography (GC or LC) interfaced to mass spectrometry (MS) has allowed for both quantitation at very low levels and unknown identification. The most revolutionary technological advances have been in MS, with significant improvements in high resolution and the overall reduction in cost of manufacturing/selling of state-of-the-art instrumentation. High resolution improvements couple with more complete libraries/databases of potential analytes, improving the ability to identify unknown contaminants while enhanced electronics have contributed to lower and lower detection limits for EDCs. These advances, in addition to improvements in sample preparation aimed at analyte isolation, have allowed analysts to measure EDC contaminants at levels approaching the newer health-based limits being evaluated by some regulatory agencies. 

In general, the instrumentation that has facilitated these improvements in detection (GC or LC-MS) share components aimed at analyte isolation and quantitation. Specifically, we think about analytes being separated from one another and from the background either chromatographically or based on their mass to charge ratio (m/z). In chromatography a mobile phase moves the sample containing multiple analytes through the stationary phase where differences in affinity of each analyte for the stationary phase causes the analytes to separate. The mass spectrometer separates the analyte mixture based on the m/z for each analyte. Additionally higher molecular weight (parent) ions can fracture/fragment into lower molecular weight (product) ions. Resolution for both chromatographic and mass spectrometric techniques is defined by the ability of each technique to separate two compounds that are very closely related in chemical characteristics. In chromatography, this usually refers to a differentiation of compounds eluting at close retention times (e.g., 1.21 vs. 1.22 min), while MS resolution refers to the differentiation of compounds with small m/z differences (e.g., 200.12 vs. 200.13). The chromatography is selected based on the compound’s volatility (boiling point), polarity, size, and even chirality. Multiclass EDC analysis needs to be broader based than analysis of any particular class of analytes (e.g., PBDEs) because there are many classes of compounds that are believed to be EDCs, with more discovered each day. 

### 9.1. Gas Chromatography Mass Spectrometry (GC-MS)

GC is used for the separation of volatile or semi-volatile compounds. Prior to injection, samples are often subjected to derivatization (i.e., with N, O-Bistrifluoroacetamide (BSTFA) or Trimethylchlorosilane (TMCS)), as many compounds require chemical modification that enable them to become GC-amenable. In a study conducted by Azzouz et al., 2014, optimization of the derivatization of 13 multiclass EDCs was required because the selected compounds were highly polar and thermally fragile; it was concluded that a mixture of BSTFA + 1% TMCS produced the most ideal signal [81]. After sample preparation, samples are carried through the GC by an inert carrier gas mobile phase (i.e., helium), where they interact with a column along with an applied temperature gradient (which heats and vaporizes the analytes), allowing for the separation of the mixture into its individual components. There are 2 main column types available for GC-instrumentation: packed columns and capillary columns. Modern research relies mostly on capillary columns, which are made of a fused silica glass tube packed with a polymer, because of their superior sensitivity and selectivity [82]. Independent from its manufacturer, the most common dimensions of a GC capillary column seen in the literature is 30 m length × 0.25 mm inner diameter × 0.25µm film thickness. After separation, the compounds are carried into the front-end source of the MS, where they are most commonly ionized by electron ionization (EI) for the creation of characteristic mass spectra. EI is known as a “hard technique” because the degree of fragmentation will destroy the m/z associated with the molecular weight of the compound (i.e., parent ion). This can present challenges when it comes to identification of unknown contaminants, which not only relies on the fragmentation information, but the molecular ions as well, which can make it challenging to match novel or under-studied compounds within a reference library. Even so, GC libraries are quite extensive; the NIST 20 mass spectral library contains over 300,000 compounds with EI spectra and 140,000 compounds with retention time information [83].

In water matrices, GC-MS is most used for the analysis of volatile or semi-volatile EDCs, such as alkylphenols, PAHs, plasticizers, and pesticides. A method developed by Martínez et al., 2013 simultaneously analyzed 76 micro pollutants, including PAHs, BPA, flame retardants, and pesticides in various river, sea, and wastewater treatment samples [84]. Following SPE treatment, chromatographic separation all compounds in 40 min was achieved using a single-quadrupole GC-MS. The method had limits of detection ranging between 0.001–0.100 ng/mL, with even better sensitivities for some non-EDC volatile organic compounds. When applied to real world samples, they only detected one EDC (tributyl phosphate) in both wastewater and river water. Azzouz & Ballesteros implemented a GC-MS method using a single-quadrupole Focus GC- DSQ II MS, analyzing 13 EDCs including parabens, phenols, BPA, and triclosan in various environmental water sources [81]. Their resulting detection limits fell in the sub-ppt range, ranging between 0.01–0.08 ng/L for all compounds, allowing for the quantification of at least one EDC compound in 87% of analyzed water samples. Another study targeted 13 EDCs (hormones, bisphenol analogues, and 4-nonylphenol) using an ion trap GC-MS, with detection limits ranging from 0.33–3.33 ng/L [85]. While simultaneously analyzing more compounds, this method proved to be more sensitive than a later study relying on a quadrupole-GC-MS analyzing only 4 EDCs (hormones, BPA, and 4-nonylphenol) whose detection limits ranged from 24.7–37 ng/mL [86]. 2 different studies utilizing LLE and no derivatization step targeted only BPA and various phthalates. They reported detection limits ranging from 0.002–0.02 [72] and 0.42–0.93 µg/L [73]. Both studies concluded that di (2-ethylhexyl) phthalate (DEHP) was the most prevalent phthalate detected in the analyzed drinking water supplies. Kumawat et al., 2022, who had the more sensitive method, had a higher frequency of detection for DEHP and DBP (87–100%) compared to Gou et al., 2016, whose reported frequency was 82% for DEHP, further highlighting the importance method sensitivity plays in the determination of EDC presence in environmental waters. However, targeted GC methods that have been used for the analysis of multi-class EDCS are overall limited in the number of compounds simultaneously analyzed. Furthermore, GC instrumentation may not be the optimal selection for the analysis of compounds in water matrices requiring derivatization prior to analysis, as many derivatization reactions are sensitive to the presence of water, impacting the intended chemical reaction. 

### 9.2. Liquid Chromatography Mass Spectrometry (LC-MS)

LC is used for the separation of polar, thermally labile compounds without the need for derivatization prior to injection [57]. Samples are carried through the LC by a liquid mobile phase composed of an aqueous solvent and an organic solvent, where they interact with a column along with an applied solvent composition gradient, allowing for the separation of the mixture into its individual components. Unlike GC, the number of column types available for LC analysis is much greater. Column selection is based on instrument pressure limitations (e.g., HPLC vs. UHPLC), affordability, and perhaps most importantly for multiclass analysis, column chemistry. The packing material lined on the inside of the column (stationary phase) is most frequently composed of hydrolyzed silica, whose functional group will determine what kind of chromatographic separation will occur within the column [87]. Typical columns that have been used previously for multiclass EDC analysis include C18 or phenyl- reverse phase columns because they typically have a broad functional pH range and can analyze compounds with varying degrees of polarity, making them great candidates for facilitating the simultaneous analysis of multiple contaminant classes. In addition to the different packing materials, column length (ranging from 15–250 mm), inner diameter (ranging from 2.1–4.6 mm), and particle size of the packing material (ranging from 1.8–5 µm) are all key dimensions that play a role in how effectively chromatographic separation of the desired analytes will occur [88]. Furthermore, there are many different components that must be optimized for chromatographic separation, including mobile phase composition, method gradient, flow rate, injection volume, and column oven temperature. After separation, the compounds are carried into the front-end source of the MS, where they are most commonly ionized by electrospray ionization (ESI) for the creation of characteristic mass spectra. ESI is known as a “soft technique” because the energy applied to a molecule is strong enough to create fragments, but weak enough to maintain the parent ion m/z. Many EDCs are considered small molecules (<1000 Da) containing a single functional group capable of carrying electrical charge [89]. When ionized by ESI, their m/z outputs can involve the loss of a proton (M − H^−^) when conducted in negative mode or the addition of a proton to the analyte (M + H^+^) in positive ion mode. Sometimes, compounds require the addition of acid or base to the mobile phase solvents to better facilitate ionization efficiency, resulting in the formation of adducts (e.g., M + Na^+^, M + formate^−^) in the final compound spectrum. In LC-MS, these small differences in conditions from one instrument to another create much larger differences in the resulting spectrum, making LC-MS spectral libraries more difficult to create in comparison to GC-MS, and less likely for an unknown analyte to fit the reference spectral data.

LC-MS is frequently chosen for the analysis of EDCs in water matrices. This platform can analyze a wide range of compounds, from polar EDCs such as phthalates to nonpolar compounds such as PBDE flame retardants, as well as compounds exhibiting both hydrophilic and hydrophobic properties like PFAS. Many classes of EDCs including PAHs, PCBs, PBDEs and others can use both GC and LC for their chromatographic separation with the smaller (lower MW analytes) perform better using GC and the larger (higher boiling point) using LC-MS exclusively. The bio-transformed (metabolized) forms of many of the actives can only be analyzed by LC-MS without derivatization as biotransformation almost always makes the analyte more polar and therefore less volatile than the original EDC. 

There are several review papers that discus the use of LC-MS platforms for individual compound classes falling under the EDC umbrella [54,55,90,91,92]. As a result of the broad capabilities of LC-MS instrumentation, many successful multiclass EDC methods have been developed and employed in the analysis of water. Anumol et al., 2013 used a 1290/6460 UHPLC-QqQ MS for the analysis of 36 organic contaminants across 7 contaminant classes, including pharmaceuticals, pesticides, steroidal hormones, personal care products, and PFAS [93]. The resulting limits of detection (0.02–10 µg/L) is indicative of a method sensitive enough to detect the trace amounts of EDC contamination anticipated to be in environmental water matrices. A more recent study published by Wee et al. in 2020 also analyzed 14 EDCs in the Malaysian drinking water supply system, covering 4 different compound classes: pharmaceuticals, hormones, and plasticizers, and pesticides [94]. In a later publication evaluating tap water from various urban housing settlements, the authors expanded their list of targeted compounds to include 18 EDCs, covering the same 4 compound classes [95]. Another method analyzed 14 EDCs in bottled water, reporting detection limits between 1.30–23.2 ng/L for phthalates, alkylphenols, BPA, and hormones [96]. However, one remaining gap in LC-MS multiclass analysis of EDCs is the small number of individual compounds included in targeted methods.

Studies covering a large range of compound classes will opt for the implementation of several different analytical methods instead. A study conducted in 2021 examining pre- and post-distribution treated waters used 14 different analytical methods to analyze for 540 unique organic contaminants, including PFAS, hormones, herbicides, and PAHs [67,97]. Several compounds (bromodichloromethane, tribromomethane, perfluorooctanoic acid) were found to be in exceedance of non-enforceable MCLGs and state advisories. While this approach allows for better optimization of each method for the individual EDC class, it requires more time and resources to complete the total analyses. Another shortcoming of many multiclass methods is the lack of focus on specific public health outcomes. González-Gaya et al., 2021 used a Dionex UltiMate 3000 UHPLC couples to a Q Exactive Focus Quadrupole-Orbitrap MS to analyze wastewater, estuary and river waters for 178 xenobiotics, only some of which exhibit endocrine disrupting properties (e.g., PFAS, BFRs) [98]. Similarly, Gago-Ferraro et al., 2020, implements a UPLC Quadrupole-Time of Flight MS method for wide-scope targeted analysis of over 2000 emerging contaminants in wastewater [69]. By designing broad-based instrumental methods for sensitive analysis of compounds with a common toxicological endpoint, quicker decisions can be made regarding chemicals whose health implications warrant fast remediation steps. 

### 9.3. High- and Low-Resolution Mass Spectrometry (HR and LRMS) 

Mass spectrometry gives us the power to identify compounds in complex matrices, either through spectral data match against a library spectrum or through accurate mass, formula weight and denovo reconstruction, or both. However, as the mixtures get more complex, it becomes increasingly important to be able to detect minute differences in m/z ratios so that differentiation of similar contaminants can be made [99]. There are several types of MS platforms that are available, and they can be separated into 2 general categories: low resolution (LR) MS and high resolution (HR) MS. The instrument preference is based on the sensitivity and selectivity requirements for the study at hand. LRMS platforms like ion-trap and single or triple-quadrupoles (QqQ) are often chosen for targeted quantification studies of complex matrices because they are highly sensitive and capable of measuring concentrations ranging from sub-ppt to ppm within the same analytical method [100]. Table 1 includes recent studies that have developed and implemented targeted methods for the simultaneous analysis of multiple classes of EDCs in various water sources utilizing LR quadrupole instruments. More advanced developments in HRMS technology, like the time-of-flight (ToF) and hybrid-quadrupole instruments like the Orbitrap, allow for the collection of accurate mass information, providing the building blocks for unequivocal identification of unknown compounds present in the matrix. 

HRMS allows for the separation of chromatographically under-resolved compounds using the parent ion differences, while lower resolution MS^n^ spectrometers allow us to distinguish compounds using fragmentation patterns. HRMS provides information to create an accurate formula weight, a significant aid in unknown identification necessary for NTA, while lower resolution MS^n^ techniques require a reconstruction of the molecule based on its fragments. Generally, this is a much tougher process, but new software packages allow for this denovo reconstruction. Use of HRMS is still not common for quantification, but for NTA, it is generally the instrument of choice. Initially, NTA was employed more frequently using GC-MS because the compound library used to identify unknowns was more uniform across instrument platforms, as the operating conditions of both the GC and MS can be matched more precisely from instrument to instrument. In other words, there are fewer moving parts—this is changing. More studies are pairing the broad-range capabilities of LC-MS with NTA methodologies, as the ability to reduce spectral libraries to their primary components, to create individual user libraries, to use MS trees (parent to product to product), as well as generating theoretical MS spectra, all facilitate identification of unknowns. There is a growing number of publications detailing a NTA approach for water contaminant analysis [103,104,105,106]. However, there are still very few publications using NTA to explicitly study environmental contaminants with common toxicological health endpoints like endocrine disruption [107,108].

It is worth noting that communication of NTA findings remains difficult because there is no standardized way of demonstrating continuity and commonality in the identification of an unknown between instruments or users. One of the more well-documented practices provides a standardized way of communicating the level of confidence we have that the identification of an unknown compound is correct [109]. Each successive level requires more information in order to confirm a compound’s identity, beginning with level 5 which needs only an accurate mass reading. Level 4 requires the determination of an unambiguous molecular formula. Levels 3 and 2 narrow in on substituent identity and plausible structures, with the scale culminating in a level 1 confirmation that requires cross checking with an analytical reference standard. There has been recent work done to help incorporate this approach, as well as provide a network of NTA databases connecting researchers globally and across industries [110].

## 10. In Vitro Biological Assays 

Commonly implemented in vitro bioassays for EDCs analyze the toxicity pathways involving estrogen, androgen, thyroid hormone receptors, as well as targets within the steroidogenesis pathway [111]. Detection using biological assays rely on mechanisms like receptor binding, cell proliferation, or luciferase induction which can then be measured and translated into a degree of activity pertinent to the compounds being studied. There are two major categorizations of in vitro bioassays as described in Campbell et al., 2006 [112]: cellular and non-cellular. Cellular bioassays rely on either yeast or human (i.e., breast cancer) cells to measure estrogenic response through the expression of a measurable protein like luciferase. Measurable responses include cell proliferation, colorimetric response, or luminescence. Non-cellular bioassays like enzyme-linked immunosorbent assays (ELISA) and enzyme-linked receptor assay (ELRA) do not require the use of whole-cells and can provide quantitative estrogenic concentration results. Additionally, there are several biosensors available that provide a hand-held, field-portable option for quick analysis of a smaller range of EDCs. A comprehensive summary on the specific types of bioassays used for EDC analysis can be found in a recent review published by Robitaille et al., 2022 [113].

Use of these biological assays allows for a sensitive and simple approach for the quick screening of water samples for endocrine disrupting activity. However, bioassays alone are not capable of discerning individual chemicals from complex mixtures like other analytical instruments. Because of this, they are often paired with GC-MS or LC-MS techniques as a way of fractionating samples, rapidly elucidating which portions of a collected water sample contain enough EDC activity to warrant further analysis into the chemical identities of those compounds causing the reaction. In a study conducted by Kakaley et al., 2021, the authors screened pre- and post-distribution treated water samples and relied on both mammalian (T47D-KBluc cell line, CV1 cell line, CV1-chAR) and yeast bioassays (BLYES, DSY-1555, DSY-105) focused on steroid hormone signaling pathways [67]. Significant bioactivity was only detected in pretreatment samples above detection limits for 2 of the tested assays. Using 14 different HPLC/MS-MS methods for organic analysis, estrone, 17α-estradiol and 17β-estradiol were quantified in the low ppt-range; however, their occurrence did not correlate with their estrogenic bioassay results. On the contrary, another study concluded that estrogen activity, which was most often detected in their analyzed environmental water samples, correlated well (r^2^ = 0.890) with 4 LC-MS measured steroidal estrogens [114]. König et al., 2017 analyzed 276 organic contaminants in surface water samples using LC-HRMS, as well as 15 cellular reporter gene bioassays [65]. Neutral fractions of the water samples were further analyzed using an HPLC coupled to a linear ion trap Orbitrap MS, and compounds deemed “important EDCs” were further analyzed using 2 additional methods performed on a hybrid triple quadrupole-linear ion trap system. These studies highlight the difficultly of relying on a single analytical method even after indicating a common mode of action using bioassays. There are still several challenges surrounding the use of bioassays for EDCs analysis. It is possible that assays are not as sensitive as the methods developed on analytical instrumentation, therefore compounds can go undetected if their concentration does not meet the threshold of biological activity selected for a particular analysis. The opposite it also plausible, where the chosen bioassay is so sensitive that the selected analytical instrument is not able to accurately measure a low enough signal. Furthermore, because of the diversity of compounds included under the category of EDCs, it is unlikely that a single biological assay will be able to detect a wide enough range of chemicals and that multiple assays would still have to be employed to effectively analyze compounds using a multiclass approach, as observed in the previously mentioned studies. 

## 11. Organ-on-a-Chip Model

Organ-on-a-chip (OoC) models are systems containing engineered or natural miniature tissues grown inside microfluidic chips [115]. These tissues can reproduce the microenvironments of human tissues and organs without the need for in vivo models. This emerging methodology has application potential for studying biological mechanisms at different complexities as small as subcellular. In the context of emerging EDCs, OoC models have the ability to predict the toxicity of chemicals in the human body more accurately than animal models, and are able to detect unanticipated biological reactions as a result of exposure [116]. However, like in vitro biological assays, OoC models alone do not have the ability to identify unknown compounds responsible for the observed biological reactions, especially when experimenting with complex mixtures. While this model is still new, there have been a few review papers published on its application for analysis of environmental xenobiotics [117,118,119]. Regarding EDC analysis, there is discussion within these papers addressing how the OoC model allows for the mimicking of endocrine signaling during pregnancy and the menstrual cycle, two critical windows of exposure for vulnerable populations. This has been previously documented in Xiao et al., 2017, whose novel OoC model allows for organ–organ integration of hormonal signaling as a representation of menstrual cycle and pregnancy-like endocrine loops, simulating the in vivo female reproductive tract [120]. 

## 12. Challenges and Future Directions

Among the challenges for evaluating any potential health risks from EDCs will be assessments with endpoints that governing bodies can agree will quantify that risk. While these are already established for cancer endpoints, no toxicant equivalency factors currently exist for EDCs; this is especially troubling for EDCs without a known or suspected cancer risk. While some activity assays are common among researchers studying EDCs (e.g., ER-α and ER-β), they are far from being labeled as the gold standard and are not currently considered for regulatory practice. Those compounds that have carcinogenic regulations (e.g., PAHs or PCBs) are regulated solely by that property while compounds such as the mycotoxin zearalenone, undergo no such concentration limit (although synthetics are regulated in the EU). Many compounds thought to be neurotoxic such as organophosphate pesticides are regulated in part based on their risk with a point of departure (LOAEL or NOAEL of cholinesterase inhibition) as the starting point for calculating a regulatory limit. While not yet proposed, this practice could also be employed for EDCs, while also recognizing many biological activity assays do not necessarily describe a negative health outcome and, like many health based regulations, may be difficult to obtain analytically. There is also a question about how to deal with multiple exposures all producing biological activity (e.g., genistein and BPA). 

In the future as more technologies become available for evaluation of potential risk and concurrently more compounds are recognized as being potential EDCs, regulatory agencies are likely to evaluate potential risk and possibly regulation based on the overall activity of a sample (e.g., water sample from a river) rather than try to tease out individual EDCs. Analytically, this would supersede the need for a multi-compound assay, either targeted or non-targeted, in favor of a biological activity assessment, although whenever there is the possibility to assign cause to a responsible party, the need for detailed chemical assays will always be necessary. EDCs are a class of emerging contaminants currently with moderate health impacts, an ever-expanding roster of contaminants, increasing numbers of assessment tools and very few regulatory limits for their endocrine activity. This will change in the future as endocrine disruption is recognized as a more important health risk, especially among the vulnerable populations of children undergoing endocrine maturation.

## Figures and Tables

**Figure 1 molecules-27-08835-f001:**
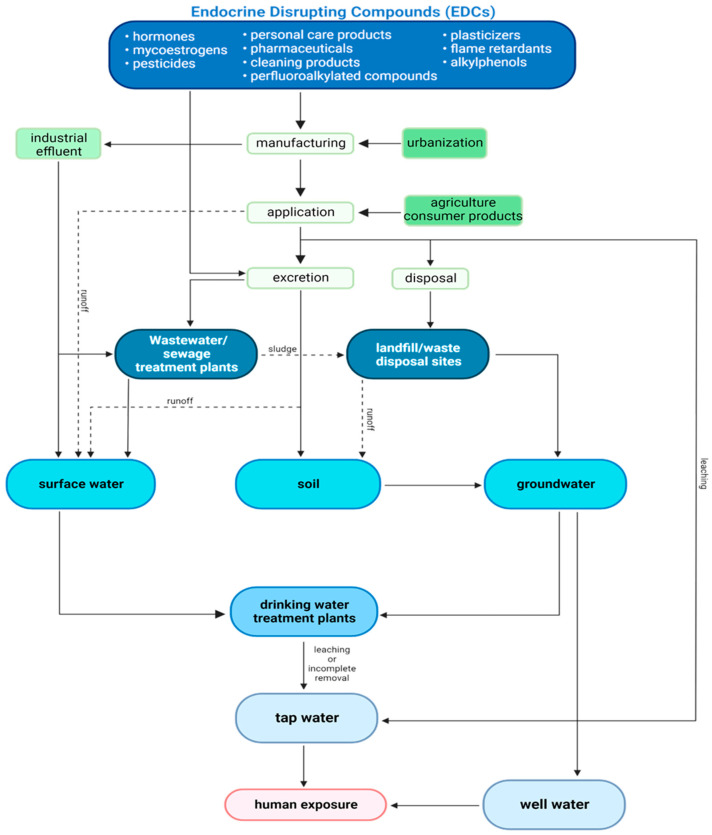
Flow chart of environmental fate and transport of EDCs within the potable water system adapted from Wee & Aris 2019 [29].

**Table 1 molecules-27-08835-t001:** Recent analytical methods for the determination of targeted multiclass endocrine disrupting compounds in different water matrices using single or QqQ MS instrumentation.

EDC Class	Water Matrix	Sample Prep	Instrumentation	Limit(s) of Detection	Reference
phthalates alkylphenols BPA hormones (n = 14)	bottled water	Micro-SPE	LC-20ADXR/LCMS-2020 HPLC-MS	1.60–23.2 ng/L	[96]
			XR-Phenyl (100 × 2.0 mm; 2.20 µm)		

phthalates, BPA (n = 7)	river water reservoir water dam water	salt-assisted LLE	Trace 1300/ISQ 7000 GC-MS TG-5MS Trace (30 m × 0.25 mm; 0.25 μm)	0.002–0.01 µg/L	[72]
hormones, pharmaceuticals, herbicides (n = 8)	drinking water	SPE	7890A/5975C GC-MS	0.02–0.087 µg/mL	[80]
			DB-35MS (30 m × 0.25 mm; 0.25 μm)		

hormones, BPA (n = 15)	wastewater influent	SPE	UHPLC-TSQ Quantiva MS	0.03–0.50 ng/L	[66]
	wastewater effluent river water tap water		Hypersil Gold C18 column (100 × 2.1 mm; 1.9 μm)		
hormones, pharmaceuticals, BPA, alkylphenols, pesticides (n = 18)	drinking water	SPE	LCMS-8030 tandem quadrupole MS Luna PFP (150 × 2.0 mm; 5 µm) Kinetex EVO C18 (150 × 2.1 mm; 5 µm)	0.01–2.56 ng/L	[95,101]
	tap water				

androgens, estrogens, glucocorticoids, progestogens (n = 23)	surface water	disk-SPE	Acquity UPLC H-class/Xevo TQ-MS QqQ Biphenyl pre-column (10 × 2.1 mm; 2.6 µm) Biphenyl column (100 × 2.1 mm; 2.6 µm)	0.035–1 ng/L	[102]


herbicides, fungicides, insecticides (n = 10)	drinking water	SPE	LC-30 UHPLC/QTRAP 6500 QqQ MS	0.01–0.64 ng/L	[64]
	bottled water		Zorbax SB-Aq (100 × 3.0 mm; 1.8 µm)		

hormones, pharmaceuticals, BPA, pesticides (n = 18)	surface water	SPE	LCMS-8030 tandem quadrupole MS	source water: 0.01–0.45 ng/L	[94]
	tap water		Luna PFP (150 × 2.0 mm; 5 µm)	tap water: 0.01–2.56 ng/L	
			Kinetex EVO C18 (150 × 2.1 mm; 5 µm)		

hormones, BPA (n = 14)	tap water surface water wastewater influent wastewater effluent	On-line SPE	UHPLC-TSQ Quantiva MS Hypersil Gold C18 (100 × 2.1 mm, 1.9 µm)	matrix-free water: 0.050–3.0 ng/L tap water: 0.10–0.70 ng/L surface water: 0.40–2.5 ng/L wastewater influent: 1.0–5.0 ng/L wastewater effluent: 0.50–4.0 ng/L	[70]
hormones, pharmaceuticals, BPA pesticides (n = 16)	surface water	SPE	LCMS-8030 tandem quadrupole MS Luna PFP (150 × 2.0 mm; 5 µm) Kinetex EVO C18 (150 × 2.1 mm; 5 µm)	0.01–0.45 ng/L	[68]

n = number of individual analytes targeted.

## Data Availability

Not applicable.

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
