# Peer review of "Recent Trends in Multiclass Analysis of Emerging Endocrine Disrupting Contaminants (EDCs) in Drinking Water"

_molecules, 2022, doi:10.3390/molecules27248835_

Round 1

Reviewer 1 Report

The manuscript “Recent Trends in Multiclass Analysis of Emerging Endocrine Disrupting Contaminants (EDCs) in Drinking Water” brings an overview of the health risks associated with exposure to EDCs mixtures, particularly those due to their presence in drinking water, together with the recent methodologies for their simultaneous determination in water samples. The authors provide the sample preparation process and pointed out the instrumentation and bioassays most widely used for the analysis of multiple classes of EDCs. From this point of view the work submitted by the authors is useful for researchers involved in this domain, and not only for them. The manuscript is well written, and the content is properly organized.

There are a few comments to be answered before acceptance:

- In section 9 (line 316) authors should discuss (briefly) the difference between resolution in chromatography and mass spectrometry.

- Line 320. One of the characteristics in the validation of an analytical method must be its specificity regardless of the number of compounds analyzed. The “less specific” term is not appropriate here.

- In section 9.1. The authors should consider adding some comment on the limitation of GC analysis compared to LC, in that it requires derivatization, extraction with a compatible organic solvent, such as dichloromethane, and in particular, avoiding the presence of water, since most derivatization reactions are sensitive to it; its presence can stop the reaction or decompose the derivatives formed.

- Sections 10 and 11. The authors should emphasize that with bioassays and OoC models is not possible to identify and quantify the EDCs (unlike LC and GC-MS); although they allow to "detect" the presence of endocrine disruptors in the sample, being especially useful for evaluating the toxicity and disruptive activity of mixtures of EDCs, but not useful for identification and quantification purposes.

Specific comments:

-Line 48-52. The authors should divide the sentence (in two, for example).

-Line 125. It is worth noting.

-Line 615, 617. [accessed 2019 November 21,]. remove comma.

-Line 782. Reference 69: write journal name without abbreviation.

Author Response

We thank the reviewers and editor for their time and effort put toward our submission to Molecules. We have addressed all the reviewers’ concerns in detail and made their suggested improvements to our manuscript to the best of our ability.

  • Special attention was given to all comments suggesting additional references to add to the manuscript narrative- we greatly appreciate the helpful recommendations.  The numbered references were adjusted accordingly.
  • Reviewer 1 (comment 1) suggested the addition of a sentence that would explain the difference between the term “resolution” within mass spectrometry and chromatography- this has been added (lines 342-345).
  • Reviewer 1 (comment 3) recommended the addition of comments surrounding the limitations of GC analysis versus LC analysis- an explicit statement on the potential interactions of derivatization agents with water, the matrix of interest, has been added to section 9.1 (lines 407-410).
  • Reviewer 1 (comment 4) suggested we emphasize the weakness of utilizing bioassay and OoC models. While we included a statement on the inability of biological assays to identify individual compounds without the aid of additional analytical instrumentation like GC or LC-MS (lines 550-554), we initially did not include a similar statement for the newer OoC models. This has been addressed within section 11, as it is critically important to remember the same weakness holds true for this novel approach (lines 601-603).

Reviewer 2 Report

The review article focuses on endocrine-disrupting compounds (EDCs). It discusses the health risks, the occurrence in the potable water system, and the current analytical protocols. I believe that the review is well-organized and can be accepted for publication after minor revisions:

Introduction: The definition of contaminants of emerging concern must be supported with a reference

Section 4: “with focus on specific emerging compounds classes such as PFAS”: Do PFAS belong to the EDCs? I know that many chemical structures have endocrine disrupting properties but I thought that PFAS do not have this property. Please, double check.

Section 5: Informal phrase “It’s”

Section 6 : "at least 2 classes of compounds qualifying as EDCs were included.”: Authors must provide their review approach. For example, which databases were searched? When and which keywords were used? How many publications were reviewed?

Section 7: “non-targeted analysis is frequently not quantitative”: There are newly developed semi-quantification approaches to address this difficulty. For example, https://link.springer.com/article/10.1007/s00216-022-04084-6

Section 8: “Goeury et al. 2019 optimized an on-line SPE protocol”: The newest approaches involve the use of multiple sorbents for SPE. This is a requirement for conducting wide-scope target screening. For example, https://www.sciencedirect.com/science/article/pii/S0304389419316668

Author Response

We thank the reviewers and editor for their time and effort put toward our submission to Molecules. We have addressed all the reviewers’ concerns in detail and made their suggested improvements to our manuscript to the best of our ability.

  • Special attention was given to all comments suggesting additional references to add to the manuscript narrative- we greatly appreciate the helpful recommendations. Additional literature sources and discussions, including a reference for the definition of contaminants of emerging concern (reviewer 2, comment 1), example(s) of semi-quantitative analysis (reviewer 2, comment 5) and SPE analysis with multiple packing materials (reviewer 2, comment 6) were added within the text and to the bibliography. The numbered references were adjusted accordingly.
  • Reviewer 2 (comment 2), “Do PFAS belong to the EDCs?” has been addressed under section 4, and explains with references, why PFAS have been included in the endocrine disruptor conversation (lines 100-104).
  • Reviewer 2 (comment 4) recommended the addition of an explanation of our review approach- an explicit description on how we found the literature sources included within the main body of the review was added (lines 210-218).